# Evaluation of Serum and Urine GDF-15 Levels in Patients with Ureteral Stones

**DOI:** 10.3390/diagnostics16010130

**Published:** 2026-01-01

**Authors:** Gorkem Akca, Ertugrul Yigit, Merve Huner Yigit, Erdem Orman, Eyup Dil, Hakki Uzun

**Affiliations:** 1Department of Urology, Faculty of Medicine, Recep Tayyip Erdogan University, Rize 53000, Türkiye; erdem.orman@erdogan.edu.tr (E.O.); eyup.dil@erdogan.edu.tr (E.D.); hakki.uzun@erdogan.edu.tr (H.U.); 2Department of Medical Biochemistry, Faculty of Medicine, Karadeniz Technical University, Trabzon 61080, Türkiye; ertugrulyigit@ktu.edu.tr; 3Department of Medical Biochemistry, Faculty of Medicine, Recep Tayyip Erdogan University, Rize 53000, Türkiye; merve.huner@erdogan.edu.tr

**Keywords:** ureteral stone, GDF-15, urinary biomarker, renal colic, inflammation

## Abstract

**Background**: Acute renal colic, most often caused by ureteral stones, is a common cause of emergency admissions. While non-contrast computed tomography (CT) is the diagnostic gold standard, its use is limited by radiation exposure, cost, and accessibility. Growth Differentiation Factor-15 (GDF-15) is a stress-induced cytokine elevated in various acute pathologies. This study investigated the diagnostic and predictive value of serum and urine GDF-15 in patients with acute renal colic due to ureteral stones. **Methods**: In this prospective observational study (January 2024–March 2025), 76 patients presenting with sudden-onset flank pain were enrolled. A total of 41 patients with radiologically confirmed ureteral stones formed the stone-positive group, and 35 patients without urinary pathology served as controls. Serum and urine GDF-15 levels were measured by ELISA, along with routine laboratory tests. CT was used to assess stone characteristics, hydronephrosis grade, and ureteral wall thickness. Group comparisons were performed using the Mann–Whitney U test, correlations with Spearman’s test, and diagnostic performance with ROC analysis. **Results**: Both serum and urine GDF-15 levels were significantly higher in stone-positive patients (*p* < 0.001). Urine GDF-15 demonstrated excellent diagnostic accuracy (AUC = 0.986; sensitivity = 92.7%; specificity = 91.4), while serum GDF-15 showed moderate performance (AUC = 0.767). GDF-15 levels showed modest positive correlations with CRP and were numerically higher in patients with ureteral wall thickness > 1 mm and proximal stones. No significant association was found with spontaneous stone passage (*p* > 0.05). **Conclusions**: Urine GDF-15 shows promising diagnostic accuracy for ureteral stones and may serve as a non-invasive adjunctive tool when imaging is limited. While associated with inflammation and stone location, it did not predict spontaneous stone passage. These findings support its potential as a clinical biomarker, though further large-scale validation is required.

## 1. Introduction

Flank pain is among the most common reasons for emergency department visits, with acute renal colic representing one of the most critical differential diagnoses. Acute renal colic typically presents as a sudden-onset flank pain originating from the costovertebral angle and radiating anteriorly and inferiorly toward the groin or testes, often accompanied by nausea and vomiting [1]. This severe pain is most commonly secondary to ureteral calculi and results from stone-induced obstruction of the ureter, leading to impaired urinary flow, ureteral spasm, and distension, ultimately stretching the renal capsule [2]. The differential diagnosis of acute renal colic should also include conditions such as myofascial pain, lumbar disc disease, acute cholecystitis, ovarian torsion, acute pancreatitis, acute diverticulitis, and acute appendicitis.

Urolithiasis is a multifactorial disease affecting approximately 5–10% of the global population, occurring about twice as frequently in men compared to women, with an increasing prevalence in recent years [3,4]. Colicky pain due to ureteral stones is typically intermittent, markedly impairs quality of life, and imposes a significant economic burden on healthcare systems [5]. In suspected ureteral stone cases, laboratory work-up generally includes urinalysis, complete blood count (CBC), C-reactive protein (CRP), and renal function tests. At the same time, non-contrast computed tomography (CT) remains the gold standard for diagnosis [6]. Although hematuria is frequently detected on urinalysis, it is absent in approximately 23% of patients [7]. Leukocytosis and elevated CRP levels may also be observed but are not specific to ureteral stones, as they may occur in various acute pathologies [8]. CT imaging, however, is limited by radiation exposure, cost, and availability. Identifying biomarkers capable of predicting ureteral stones could reduce unnecessary CT use, thereby improving both patient safety and healthcare system efficiency.

Growth Differentiation Factor-15 (GDF-15) is a stress-response cytokine belonging to the transforming growth factor-beta (TGF-β) superfamily, whose expression increases in response to various pathophysiological stimuli, including cellular inflammation, tissue injury, hypoxia, and oxidative stress [9]. Although GDF-15 circulates at low levels under normal physiological conditions, its serum concentration rises markedly in numerous chronic diseases—such as cardiovascular disorders, chronic kidney disease, malignancies, and infectious conditions—as well as in acute pathologies including myocardial infarction, pulmonary embolism, and trauma, similar to other acute-phase reactants [10,11]. Studies focusing on the urogenital system have shown that GDF-15 plays a role particularly in tubulointerstitial injury and inflammatory processes, supporting its potential utility as a renal stress and injury biomarker [12,13]. Therefore, its potential as a biomarker in renal colic secondary to ureteral stones warrants investigation.

The spontaneous passage of ureteral stones varies depending on factors such as stone size, location, surrounding edema, inflammation, and ureteral motility. In recent years, there has been growing interest in exploring biochemical markers that may influence this process. Given its involvement in inflammatory responses and tissue injury, GDF-15 emerges as a promising candidate in this regard. However, no studies to date have directly examined the relationship between GDF-15 levels and ureteral stone passage. Investigating the predictive potential of GDF-15 as a biomarker in this context could offer a novel perspective in the clinical management of ureteral stones.

This study was designed to investigate the diagnostic and predictive value of serum and urine GDF-15 levels in patients with acute renal colic due to ureteral stones. In addition, it aims to evaluate the potential utility of GDF-15 in predicting the spontaneous passage of ureteral stones measuring 2–10 mm, and to assess its association with other predictive factors such as ureteral wall thickness, serum CRP, and hematological parameters. This approach may help establish the clinical applicability of GDF-15 in differential diagnosis and treatment planning, particularly in patients presenting to the emergency department with sudden-onset flank pain who are scheduled for imaging.

## 2. Materials and Methods

### 2.1. Ethical Approval, Study Design, and Patient Selection

This prospective observational study was approved by the Scientific Research Ethics Committee of Rize Recep Tayyip Erdoğan University Faculty of Medicine (Approval Date: 4 January 2024; Decision No: 2024/03). The study was conducted following the principles of the Declaration of Helsinki and relevant national regulations. Between January 2024 and March 2025, patients who presented to our Urology outpatient clinic with sudden-onset flank pain were evaluated for eligibility. Written informed consent was obtained from all participants before enrollment. Patients were divided into two groups. Those diagnosed with ureteral stones based on physical examination, blood and urine tests, and computed tomography (CT) imaging were assigned to the stone-positive group (*n* = 41). Patients in whom ureteral stones or any other urinary tract pathology were not detected were designated as the stone-negative control group (*n* = 35). All examinations, diagnostic procedures, treatments, and follow-ups were conducted by three urologists (GA, HU, and ED). Demographic characteristics, history of urolithiasis, detailed medical history, and physical examination findings were recorded at the time of admission. Inclusion criteria were: age between 20 and 65 years, presence of a single ureteral stone measuring 2–10 mm in any location, and absence of active urinary tract infection. Patients with a solitary kidney, those diagnosed with acute pyelonephritis secondary to ureteral stones, patients with ureteral stones < 2 mm or >10 mm, individuals with multiple ureteral stones, and pregnant women were excluded. A total of 76 patients who met the inclusion criteria were enrolled: 41 in the stone-positive group and 35 in the stone-negative group. Sample size calculation was performed using G*Power version 3.1. Assuming a moderate effect size (d = 0.60), a significance level (α) of 0.05, and a statistical power (1−β) of 0.90, the required total sample size was calculated to be 72 participants (36 per group). The selected effect size was based on previously published clinical studies evaluating urinary and serum GDF-15 as renal and inflammatory biomarkers in patient cohorts of comparable size, in which moderate between-group differences were reported [12,13]. Therefore, including 76 participants (*n* = 41 vs. *n* = 35) was deemed sufficient to meet statistical power requirements.

### 2.2. Blood and Urine Sample Collection and Biochemical Parameter Measurements

Upon admission to the emergency department with acute flank pain suggestive of renal colic, all patients underwent routine laboratory evaluations, including measurements of C-reactive protein (CRP), serum urea, and creatinine levels. Venous blood samples were collected between 08:00 and 11:00 a.m., preferably during the acute episode and before the initiation of analgesic or medical expulsive therapy. For patients presenting later—within 24–72 h after symptom onset—samples were collected at the time of admission, and the time interval between symptom onset and sampling was recorded. All analyses were performed in the Biochemistry Laboratory of our hospital. CRP, blood urea nitrogen (BUN), and serum creatinine levels were measured using the Beckman Coulter AU5800 fully automated biochemistry analyzer (Brea, CA, USA). The estimated glomerular filtration rate (eGFR) was calculated using the modified MDRD formula based on serum creatinine values. Complete blood count parameters—including white blood cell (WBC) count, neutrophil (Neu) count, lymphocyte (Lym) count, and hemoglobin (Hb)—were determined with the Mindray BC-6000 fully automated hematology analyzer (Shenzhen, China). The neutrophil-to-lymphocyte ratio (NLR) was calculated as an additional inflammation-related parameter. Following the routine laboratory tests, serum and urine samples were obtained for GDF-15 measurement. These samples were stored at −80 °C until analysis.

### 2.3. Imaging Method

Following clinical and laboratory assessments, non-contrast-enhanced whole-abdomen computed tomography (CT) was performed for radiological evaluation. The scans were obtained using a 64-multidetector CT scanner (Discovery CT750 HD, GE Healthcare, Chicago, IL, USA) with a collimated slice thickness of 0.625 mm at the isocenter and an operating voltage of 120 kVp. This imaging modality was used to determine the location of the ureteral stone, its longitudinal and transverse dimensions, the presence and grade of proximal hydroureteronephrosis, the stone area, and ureteral wall thickness (UWT) at the level of the stone. The stone location was categorized as proximal, mid, or distal ureter. Proximal ureteral stones were defined as those located between the ureteropelvic junction (UPJ) and the upper border of the sacrum, mid-ureteral stones as those at the level of the sacrum, and distal ureteral stones as those between the lower border of the sacrum and the ureteral orifice [14]. The size of each stone was measured in the coronal plane using its maximum longitudinal and perpendicular diameters. The stone area was calculated by multiplying these two dimensions [15]. Hydronephrosis severity was assessed according to a standardized grading system and recorded as mild, moderate, or severe. Mild hydronephrosis was defined as dilatation of the renal pelvis and a few calyces; moderate as dilatation of the renal pelvis and all calyces; and severe as hydronephrosis accompanied by renal parenchymal atrophy [16]. Ureteral wall thickness (UWT) was measured on axial slices by assessing the maximal soft tissue thickness surrounding the stone at the level of its localization [17].

### 2.4. Follow-Up Protocol

Following diagnostic evaluation, patients diagnosed with ureteral stones were managed conservatively for two weeks. Conservative management included medical expulsive therapy (MET), ensuring adequate fluid intake, and supportive care. MET consisted of a once-daily alpha-blocker along with analgesics when needed. Patients were re-evaluated after two weeks to determine whether spontaneous stone passage had occurred. For those who had not passed their stones, alternative treatment options such as ureterorenoscopy (URS) or extracorporeal shock wave lithotripsy (ESWL) were offered. Patients who required surgical intervention during the follow-up period were excluded from the analysis of spontaneous stone passage outcomes, as conservative management could no longer be evaluated in these cases.

### 2.5. Measurement of Serum and Urine GDF-15 Levels

Following centrifugation, serum and urine samples were aliquoted into single-use tubes and stored at −80 °C until the target sample size was reached. All samples were then thawed, and GDF-15 levels were measured. The measurements were performed in a double-blind manner. GDF-15 concentrations were determined using a commercially available ELISA kit (R&D Systems, DGD150, Minneapolis, MN, USA), in accordance with the manufacturer’s protocol.

### 2.6. Statistical Analysis

Statistical analyses were performed using OriginPro 2025 software (OriginLab Corporation, Northampton, MA, USA). The distribution of continuous variables was assessed using the Shapiro–Wilk test. Non-normally distributed data were expressed as median (Min–Max), and differences between groups were analyzed using the Mann–Whitney U test. Nominal variables were presented as frequencies and percentages (%). Spearman’s rank correlation analysis was used to assess the relationship between serum and urine GDF-15 levels and various biochemical and hematological parameters. A *p*-value of <0.05 was considered statistically significant. Additionally, ROC (Receiver Operating Characteristic) curve analysis was performed to evaluate the diagnostic utility of serum and urine GDF-15 levels. The area under the curve (AUC), sensitivity, and specificity were calculated. Optimal cut-off points were determined according to the Youden index.

## 3. Results

### 3.1. Comparison of Demographic and Clinical Features Between Stone-Negative and Stone-Positive Patients

A comparison of demographic and clinical characteristics between patients with and without ureteral stones is presented in Table 1. Although age and weight values were similar between the groups (*p* > 0.05), the stone-positive group had significantly greater height (*p* < 0.05). The frequency of smoking was also considerably higher among stone-positive patients (51.2%) compared to the stone-negative group (25.7%) (*p* < 0.05). The distribution of pain duration did not differ markedly between groups.

### 3.2. Serum and Urine GDF-15 Levels in Patients with and Without Ureteral Stones

Serum and urine GDF-15 levels were significantly higher in patients with radiologically confirmed ureteral stones (stone-positive, *n* = 41) compared to those without urinary tract pathology (stone-negative, *n* = 35) (both *p* < 0.001). Specifically, serum GDF-15 concentrations were elevated in the stone-positive group (Figure 1A), while urine GDF-15 levels showed a similar pattern (Figure 1B). These findings support the potential utility of GDF-15, in both serum and urine, as a biomarker for the presence of ureteral stones.

### 3.3. Inflammatory and Hematologic Parameters in Patients with and Without Ureteral Stones

Serum CRP levels were significantly higher in the stone-positive group compared to stone-negative controls (*p* = 0.014), indicating an acute-phase inflammatory response (Figure 2A). In contrast, WBC and Neu counts, although numerically higher in the stone-positive group, did not reach statistical significance (*p* = 0.054 and *p* = 0.062, respectively) (Figure 2B,C). This lack of a statistically significant difference in leukocyte counts may be partly explained by the fact that blood samples were collected not only during the acute renal colic episode but also from some patients presenting within 24–72 h after symptom onset, when systemic inflammatory responses might have partially subsided. Lym counts and NLR were similar between groups (*p* = 0.478 and *p* = 0.199, respectively) (Figure 2D,E). Hb levels were also comparable (*p* = 0.577), suggesting no acute anemia or bleeding tendency associated with stone status (Figure 2F). These findings indicate that, in this cohort, CRP may serve as a more sensitive supportive marker of inflammation in suspected ureteral stone cases than hemogram-derived indices.

### 3.4. Evaluation of Kidney Function Biomarkers in Patients with and Without Ureteral Stones

To assess the impact of ureteral stones on renal function, serum BUN, creatinine, and eGFR were compared between groups. Although BUN levels were slightly higher in stone-positive patients, the difference was not statistically significant (*p* = 0.151) (Figure 3A). Serum creatinine levels were significantly elevated in the stone-positive group compared to stone-negative controls (*p* = 0.025) (Figure 3B), and correspondingly, eGFR values were significantly lower in stone-positive patients (*p* = 0.035) (Figure 3C). These findings suggest that even in the acute setting, the presence of ureteral stones may transiently impair renal function, as reflected by changes in serum creatinine and eGFR.

### 3.5. Correlation of Serum and Urine GDF-15 Levels with Other Parameters

Spearman correlation analysis was performed to evaluate the relationship between serum and urine GDF-15 levels and various hematological and biochemical parameters (Figure 4). A significant but mild-to-moderate positive correlation was found between serum GDF-15 levels and CRP (r = 0.385, *p* < 0.05). Urine GDF-15 levels demonstrated a weak but statistically significant correlation with CRP (r = 0.275, *p* < 0.05), indicating a limited association with systemic inflammatory activity. In addition, a moderate and statistically significant correlation was observed between serum and urine GDF-15 levels (r = 0.427, *p* < 0.05), indicating consistency of GDF-15 concentrations across the two biological fluids. However, correlations between GDF-15 and other parameters were mostly weak and not clinically significant. These findings suggest that GDF-15 is partially associated with inflammation-related markers and demonstrates similar trends in both serum and urine.

### 3.6. Evaluation of the Diagnostic Performance of Serum and Urine GDF-15 Levels Using ROC Analysis

ROC curve analysis was performed to evaluate the diagnostic performance of GDF-15 levels for identifying ureteral stones. The area under the curve (AUC) for urinary GDF-15 was 0.986 (95% CI: 0.967–1.000; *p* < 0.0001), indicating excellent diagnostic accuracy (Figure 5). At the optimal cut-off value of 438.3 pg/mL, urinary GDF-15 demonstrated a sensitivity of 92.7% (95% CI: 80.1–98.5%) and a specificity of 91.4% (95% CI: 76.9–98.2%).

For serum GDF-15, the AUC was 0.767 (95% CI: 0.660–0.873; *p* < 0.0001), reflecting moderate diagnostic performance. Using the Youden index, the optimal serum GDF-15 cut-off value was 537.15 pg/mL, which yielded a sensitivity of 43.9% (95% CI: 28.5–60.3%) and a specificity of 100% (95% CI: 89.9–100%).

Overall, these results indicate that both serum and urinary GDF-15 levels are associated with the presence of ureteral stones, with urinary GDF-15 demonstrating superior diagnostic accuracy. This finding supports the potential utility of urinary GDF-15 as a non-invasive adjunctive biomarker in patients presenting with acute flank pain.

### 3.7. Radiological Characteristics and GDF-15 Levels

Urine and serum GDF-15 levels demonstrated variation according to stone location, hydronephrosis grade, and ureteral wall thickness (Table 2). Median urine GDF-15 concentrations were higher in proximal ureter stones (1903 pg/mL) compared to mid (1757 pg/mL) and distal (1529 pg/mL) locations, with serum GDF-15 showing a similar trend, being highest in proximal stones (472.8 pg/mL); however, these differences did not reach statistical significance (*p* > 0.05 for all comparisons). Patients with ureteral wall thickness greater than 1 mm exhibited numerically higher GDF-15 levels in both urine (2171 pg/mL) and serum (432.5 pg/mL) compared to those with ≤1 mm thickness; however, these differences were not statistically significant (*p* > 0.05). Although an apparent trend toward lower GDF-15 concentrations with increasing hydronephrosis grade was observed (Grade 0: urine 2167 pg/mL, serum 537.1 pg/mL; Grade 1: urine 1811 pg/mL, serum 468.8 pg/mL; Grade 2: urine 1637 pg/mL, serum 334.8 pg/mL), these differences were not statistically significant (*p* > 0.05). This finding is counterintuitive to the expected pathophysiology, as hydronephrosis—often associated with renal functional impairment—would be anticipated to increase GDF-15 levels. The observed discrepancy may be attributable to differences in timing of sample collection relative to the acute obstruction episode, variations in baseline renal function, or the limited sample size within hydronephrosis subgroups.

## 4. Discussion

This study investigated the potential utility of serum and urine GDF-15 levels in diagnosing ureteral stones and predicting spontaneous stone passage in patients presenting with acute flank pain. Our findings demonstrated that both serum and urine GDF-15 levels were significantly higher in individuals with ureteral stones compared to those without. Notably, urine GDF-15 levels exhibited strong diagnostic performance, with high sensitivity and specificity for detecting stone presence (AUC: 0.986), whereas serum GDF-15 showed moderate diagnostic accuracy. The urine GDF-15 cut-off value of 438.3 pg/mL identified in this study represents a data-driven threshold derived from the current cohort. Although it demonstrated excellent diagnostic performance, this value should be considered exploratory, as optimal cut-off levels may vary according to patient characteristics, assay platforms, and clinical settings. External validation in independent cohorts is therefore required before routine clinical implementation. In addition, both serum and urine GDF-15 levels were positively correlated with CRP and were higher in patients with ureteral wall thickness >1 mm and in those with proximally located stones. However, no statistically significant association was found between GDF-15 levels and spontaneous stone passage. These results suggest that GDF-15 may increase in response to renal stress and inflammatory processes associated with ureteral obstruction; however, the observed correlations with CRP were modest, indicating that GDF-15 is unlikely to function as a direct surrogate marker of systemic inflammation.

Although non-contrast computed tomography (CT) remains the diagnostic gold standard for ureteral stones, its widespread use in emergency settings is associated with radiation exposure, cost, and limited accessibility. In addition, the nonspecific clinical presentation of acute flank pain may lead to unnecessary imaging. Previous studies have shown that a substantial proportion of CT examinations performed for suspected renal colic are clinically unnecessary, highlighting concerns regarding CT overuse in emergency departments. These limitations underscore the need for supportive diagnostic tools that may aid clinical decision-making, particularly when imaging is delayed or not readily available. In this context, biomarkers such as GDF-15 may provide complementary diagnostic information rather than replace imaging [18]. GDF-15 is a cytokine known to be upregulated in response to cellular stress and tissue injury, with established prognostic and diagnostic value in various conditions, including cardiovascular disease, malignancy, and kidney disease [9,10,11,12]. In the renal context, GDF-15 has been reported to reflect tubulointerstitial injury [12,13]. Ureteral stones are characterized by local inflammation, edema, and alterations in peristaltic activity. In our study, the significant correlation between GDF-15 levels and ureteral wall thickness suggests that this biomarker may reflect localized inflammatory responses. Furthermore, the positive correlation between CRP and GDF-15 supports the notion that GDF-15 is also sensitive to systemic inflammatory activity. Although CRP has been reported to have limited predictive value for stone passage [8], its low specificity limits its utility as a stand-alone diagnostic or prognostic marker. By contrast, the high accuracy of urine GDF-15 makes it a more promising candidate in the differential diagnosis of acute flank pain. Experimental studies have further characterized GDF-15 as a stress-responsive cytokine involved in tissue tolerance during inflammatory states, contributing to protection against inflammation-induced organ injury [19,20,21]. In this context, elevated GDF-15 levels in patients with ureteral stones may represent a compensatory response to acute renal stress and inflammation rather than a stone-specific inflammatory signal.

Supporting this interpretation, our previous prospective study in patients with urosepsis demonstrated that serum GDF-15 levels decreased significantly during clinical recovery following the acute inflammatory phase. In that cohort, reductions in GDF-15 occurred in parallel with improvements in inflammatory markers and renal function parameters, suggesting that GDF-15 reflects dynamic changes associated with the resolution of systemic inflammatory stress rather than representing a static disease-specific marker [22].

Our observation that serum and urine GDF-15 levels were significantly higher in patients with ureteral stones compared to controls is supported by evidence from both experimental and clinical studies. Sawasawa et al. demonstrated that serum GDF-15 levels rise early in obstructive kidney injury, indicating its potential as an early biomarker of proximal tubular cell damage [23]. Furthermore, Farag et al. reported that GDF-15 levels were significantly higher in chronic kidney disease patients compared to healthy controls, supporting the association between GDF-15 elevation and renal dysfunction across diverse etiologies [24]. Consistent with these findings, our recent clinical study in gouty arthritis patients showed markedly elevated serum GDF-15 concentrations, particularly during acute attacks, and a significant correlation with impaired renal function parameters, underscoring its role as a marker of both inflammation and kidney injury in systemic inflammatory conditions [25]. Collectively, these findings reinforce the interpretation that elevated GDF-15 in ureteral stone cases likely reflects the presence and severity of obstruction-induced renal insult.

The superior diagnostic performance of urine GDF-15 observed in our study is in line with previous research highlighting its potential advantages over serum measurements. Perez-Gomez et al. demonstrated that while plasma GDF-15 levels are influenced by renal impairment, urine GDF-15 more accurately reflects underlying kidney pathology (histological injury) and independently predicts renal function decline in chronic kidney disease [14]. In a similar context, Oshita et al. reported that urine GDF-15 levels were less dependent on glomerular filtration rate and predicted CKD progression with an accuracy comparable to albuminuria [13]. These findings suggest that urine GDF-15 may provide a more direct signal of renal tissue injury, potentially offering an advantage in conditions like ureteral obstruction where localized damage is present. Although stone-specific ROC data for GDF-15 are lacking in the literature, analogous studies in CKD and sepsis have reported high diagnostic accuracy for GDF-15, supporting the plausibility of our findings and reinforcing its potential clinical value in acute obstructive urologic settings [26].

Most existing literature on GDF-15 has focused on serum measurements and their associations with various systemic diseases. Studies on urine GDF-15 are scarce, and data on its use in diagnosing and differentiating acute obstructive uropathies such as ureteral stones are lacking. Our study addresses this gap, showing that urine GDF-15 measurement could serve as a non-invasive, rapid diagnostic tool for detecting ureteral stones. Considering the ease of urine collection, assessment of urine GDF-15 could provide a practical contribution to diagnostic workflows in emergency departments. This positions our work as one of the first to provide evidence supporting the clinical use of GDF-15 in urine as well as in serum. Differences in GDF-15 levels by stone location, with higher values in proximal stones, although these differences did not reach statistical significance, may be explained by increased obstruction severity and elevated intrarenal pressure. This finding suggests that GDF-15 may be influenced not only by inflammation but also by secondary changes such as hydronephrosis. On the other hand, the absence of a statistically significant relationship between GDF-15 levels and spontaneous stone passage may reflect the multifactorial nature of stone passage, which depends on factors beyond inflammation or ureteral wall thickness.

This study has certain limitations. The relatively small sample size and single-center design may limit generalizability. In addition, GDF-15 is known to be elevated in a wide range of systemic conditions, including cardiovascular disease, chronic kidney disease, and malignancy, reflecting its role as a general stress-responsive cytokine rather than a disease-specific marker. This characteristic may limit its specificity in emergency settings, particularly in patients with multiple comorbidities, and underscores the need to interpret GDF-15 levels within the broader clinical context rather than as a standalone diagnostic tool. Cardiovascular or metabolic comorbidities that could influence GDF-15 levels may not have been fully excluded. Given that GDF-15 can increase in various renal and systemic conditions, future studies should aim to control for such confounders. Moreover, commonly used treatments in renal colic management, such as analgesics and medical expulsive therapy (MET) agents, including alpha-blockers, may influence inflammatory pathways and systemic stress responses. Although these treatments were administered according to standard clinical practice, their potential effects on inflammatory biomarkers cannot be entirely excluded. The two-week follow-up period for assessing stone passage may also have overlooked longer-term outcomes. In addition, patients who had already passed their ureteral stones prior to presentation but exhibited residual inflammatory findings were not included as a separate group; therefore, potential differences in GDF-15 levels between active ureteral obstruction and post-obstructive residual inflammation could not be evaluated in the present study. Finally, pre-analytical factors such as technical variability in GDF-15 assay kits or differences in sample collection timing could have influenced the results. However, although careful attention was paid to standardized sample handling and storage procedures in order to minimize pre-analytical variability, the use of freshly processed samples might provide even more accurate measurements of GDF-15. Given that GDF-15 is not routinely measured in acute clinical settings, frozen sample analysis represents a practical and widely accepted approach in biomarker studies.

## 5. Conclusions

This study demonstrated that both serum and urine GDF-15 levels were significantly elevated in patients with radiologically confirmed ureteral stones compared with individuals without urinary tract pathology. Notably, urine GDF-15 exhibited excellent diagnostic performance, suggesting its potential use as a non-invasive biomarker in the evaluation of acute flank pain. Serum GDF-15 levels showed moderate predictive value for stone presence, and their positive correlation with inflammatory markers such as CRP supports the notion that GDF-15 may reflect both local and systemic inflammatory responses associated with ureteral obstruction. Our work is among the first to measure urine GDF-15 and associate it with ureterolithiasis, indicating a novel and practical application for this biomarker in clinical urology. Although no statistically significant association was found between GDF-15 levels and spontaneous stone passage, GDF-15 tended to be higher with inflammatory markers (e.g., ureteral wall thickness), but this did not reach statistical significance. Importantly, urine GDF-15 was not identified as a predictor of spontaneous stone passage in the present study, underscoring this finding as a meaningful negative result. These findings suggest that urine GDF-15 measurement could be integrated as a supportive diagnostic tool, particularly in patients for whom imaging modalities are not immediately accessible or are delayed. Further large-scale studies with more extended follow-up periods are warranted to better elucidate the prognostic value of GDF-15 in predicting clinical outcomes and guiding treatment strategies.

## Figures and Tables

**Figure 1 diagnostics-16-00130-f001:**
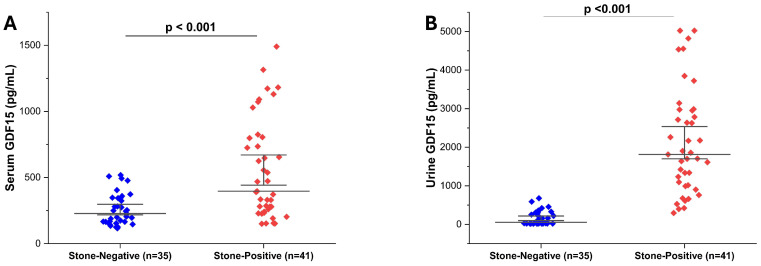
Comparison of serum and urine GDF-15 levels between stone-positive and stone-negative patients. (**A**) Serum GDF-15 levels (pg/mL) were significantly higher in patients with radiologically confirmed ureteral stones (*n* = 41) than in those without evidence of stone (*n* = 35). (**B**) Similarly, urine GDF-15 concentrations were significantly elevated in the stone-positive group. Data are presented as individual values with mean ± standard deviation. Statistical analysis was performed using the Mann–Whitney U test due to the non-normal distribution of the data. *p* < 0.001 was considered statistically significant.

**Figure 2 diagnostics-16-00130-f002:**
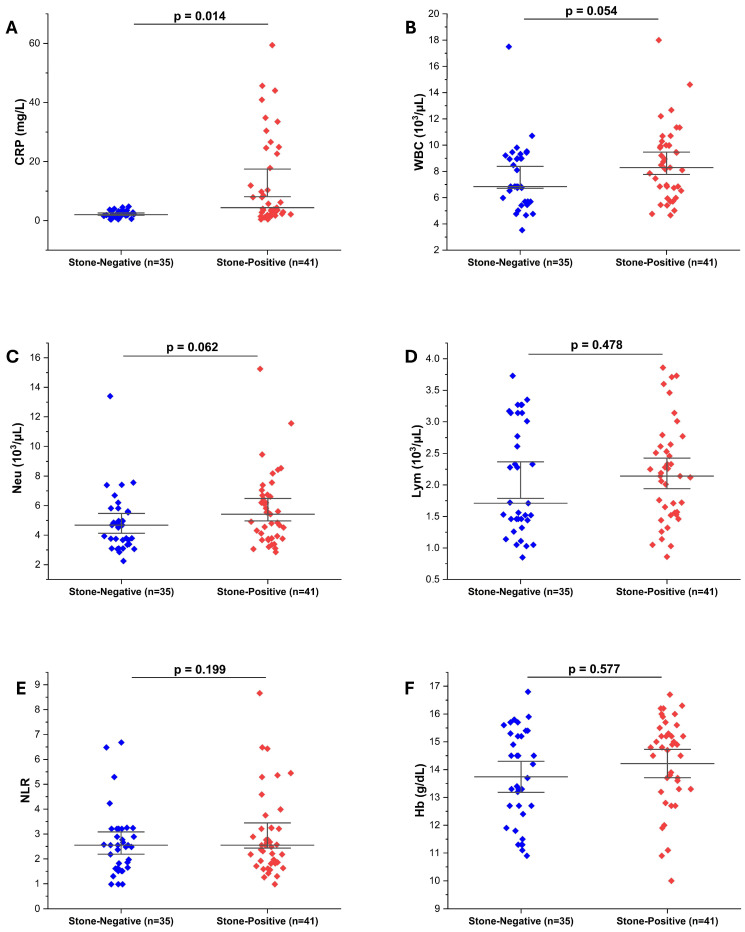
Comparison of inflammatory and hematologic biomarkers in stone-negative and stone-positive patients. (**A**) CRP levels were significantly higher in stone-positive patients (*p* = 0.014). (**B**) WBC counts showed a borderline increase in the stone-positive group (*p* = 0.054). (**C**) Neutrophil counts showed a non-significant upward trend (*p* = 0.062). (**D**) Lymphocyte counts did not differ significantly (*p* = 0.478). (**E**) NLR values were also comparable between groups (*p* = 0.199). (**F**) Hemoglobin levels showed no significant difference (*p* = 0.577). Statistical comparisons were made using the Mann–Whitney U test.

**Figure 3 diagnostics-16-00130-f003:**
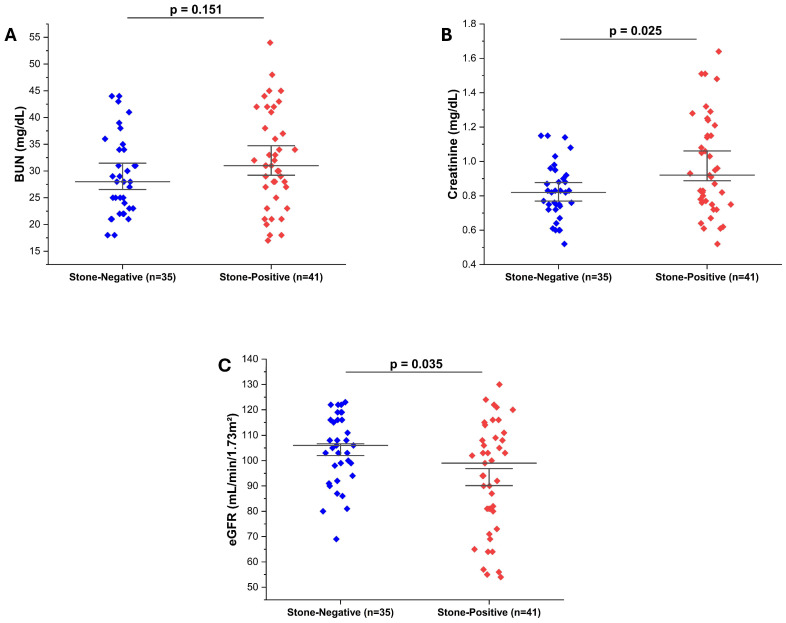
Kidney function biomarkers in stone-negative and stone-positive patients. (**A**) Blood urea nitrogen (BUN) levels showed no significant difference between groups (*p* = 0.151). (**B**) Serum creatinine levels were significantly higher in stone-positive patients (*p* = 0.025). (**C**) eGFR values were significantly lower in the stone-positive group (*p* = 0.035), suggesting a mild reduction in renal function. Group comparisons were performed using the Mann–Whitney U test.

**Figure 4 diagnostics-16-00130-f004:**
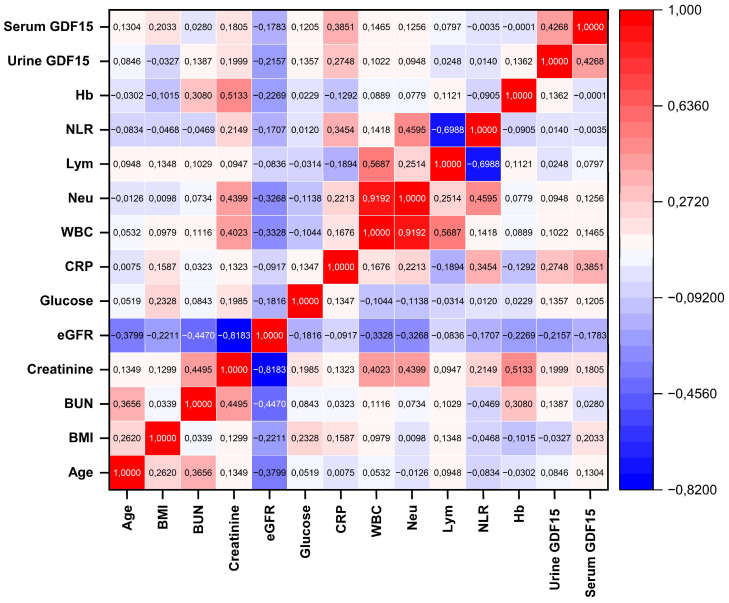
Heatmap representation of Spearman correlations between serum and urine GDF-15 levels and hematological and biochemical parameters.

**Figure 5 diagnostics-16-00130-f005:**
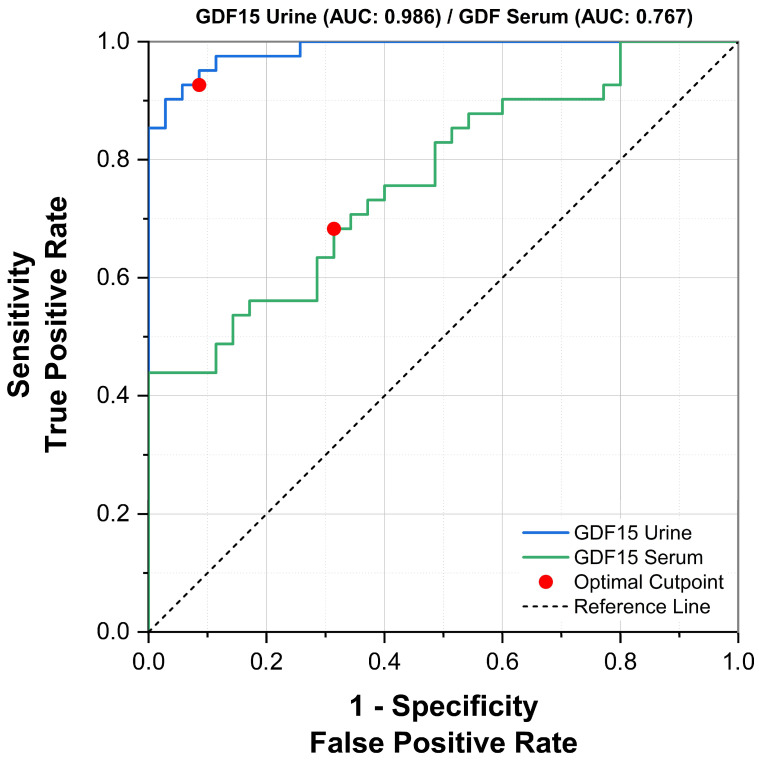
Receiver operating characteristic (ROC) curves of serum and urinary GDF-15 levels for predicting the presence of ureteral stones.

**Table 1 diagnostics-16-00130-t001:** Comparison of demographic and clinical characteristics between stone-negative and stone-positive patients.

	Stone-Negative (*n* = 35) Median (Min–Max)	Stone-Positive (*n* = 41) Median (Min–Max)	*p* (* Mann–Whitney U † Chi-Square)
Sex			
[*n* (%) Female,	20 (57.1%),	22 (53.6%)	† *p* > 0.05
*n* (%) Male]	15 (42.9%)	19 (46.4%)	
Age (Years)	41 (22–52)	41 (21–65)	* *p* > 0.05
Height (cm)	163 (150–181)	170 (149–196)	* *p* < 0.05
Weight (kg)	76 (54–117)	82 (44–110)	* *p* > 0.05
BMI (kg/m^2^)	27.8 (18.9–46.6)	27.7 (18.1–35.9)	* *p* > 0.05
Smoke [*n* (%)]	9 (25.7%)	21 (51.2%)	† *p* < 0.05
Alcohol [*n* (%)]	2 (5.7%)	3 (7.3%)	
Medical History of Ureteral Stone (*n*)	10 (28.5%)	21 (51.2%)	
Pain Duration			
(Two Days, *n*)	3	3	
(Three Days, *n*)	10	16	
(Four Days, *n*)	22	22	
BT Findings			
[Negative, *n* (%)]	35 (100%)	0 (0%)	
[Positive, *n* (%)]	0 (0%)	41 (100%)	

* Mann–Whitney U. † Chi-square.

**Table 2 diagnostics-16-00130-t002:** Distribution of Urine and Serum GDF-15 Levels According to Radiological Findings and Clinical Outcomes in Stone-Positive Patients.

	Number	* Urine GDF-15 [Median (Min–Max)], † Spearman Correlation	* SerumGDF-15 [Median (Min–Max)], † Spearman Correlation
Stone Location			
Distal, *n* (%)	14 (34%)	1529 (422.1–5022) *	308.1 (149.8–3130) *
Mid, *n* (%)	6 (15%)	1757 (527.4–3722) *	381.2 (151.8–1490) *
Proximal, *n* (%)	21 (51%)	1903 (293.8–5022) *	472.8 (152.7–1172) *
Side			
Right, *n* (%)	20 (49%)	1833 (293.1–5022) *	432.5 (152.7–3130) *
Left, *n* (%)	21 (51%)	1698 (395.8–4820) *	390.7 (149.8–1314) *
Hydronephrosis Grade			
Grade 0, *n* (%)	2 (4.8%)		
Grade 1, *n* (%)	16 (39%)	2167 (385.8–4820) *	537.1 (149.8–1490) *
Grade 2, *n* (%)	16 (39%)	1811 (293.8–5022) *	468.8 (191.1–3130) *
Grade 3, *n* (%)	7 (17%)	1637 (610.4–3848) *	334.8 (228.9–647.1) *
Ureteral Wall Thickness			
≤1 mm, *n* (%)	11 (29%)	1094 (395.8–3722) *	390.7 (151.8–1181) *
>1 mm, *n* (%)	30 (71%)	2171 (293.8–5022) *	432.5 (149.8–3130) *
		r = 0.09, *p* > 0.05 †	r = 0.089, *p* > 0.05 †
Stone Area (mm^2^)			
Clinical Outcome			
Spontaneous passage	23 (56.1%)	1703 (395.8–5022) *	396.1 (149.8–3130) *
URS performed	15 (36.5%)	1811 (293.8–5004) *	468.8 (203–1172) *
ESWL performed	3 (7.4%)	2626 (659–2635) *	329.5 (152.7–472.8) *

* GDF-15 [Median (Min–Max)], † Spearman Correlation.

## Data Availability

The original contributions presented in this study are included in the article; further inquiries can be directed to the corresponding author.

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
