# Peer review of "Evaluation of Serum and Urine GDF-15 Levels in Patients with Ureteral Stones"

_diagnostics, 2026, doi:10.3390/diagnostics16010130_

Round 1
Reviewer 1 Report
Comments and Suggestions for Authors
The study is generally well-designed, with appropriate statistical methods and clear reporting. To improve the manuscrip a few changes are necessary.
1. Although a power calculation is referenced (G*Power; d = 0.60; α = 0.05; power = 0.90), the chosen effect size is not justified. Please cite precedent studies or provide a rationale for selecting d = 0.60.
Despite the link between serum GDF-15 and CRP being characterized as 'significant' (r = 0.385, p < 0.05), its strength is mild to moderate. This should be reframed to avoid overstatemet.
Similarly, there is virtually little connection (r = 0.275) between urine GDF-15 and CRP. The wording should properly acknowledge this.
2. The discussion is lengthy and repetitive in places. There are several pages devoted to CT overuse, for example. It should be condensed and rearranged.
3. Lines 358–367, which describe GDF-15 as a "inflammation-induced central mediator," seem a little pointless. It should be integrated more concisely.
4. Describe how GDF-15 is raised in a number of systemic illnesses, including cancer, chronic kidney disease (CKD), and cardiovascular disease (CVD). What impact might this have on specificity in actual emergency situations?.
Consider how analgesics or MET (alpha-blockers) may affect inflammatory indicators.
5. Concerning the ROC Analysis Presentation, give sensitivity and specificity 95% confidence intervals (CIs) in addition to the area under the curve (AUC).
The cut-off point for urinary GDF-15 (438.3 pg/mL) should be discussed.
6. Declare clearly in the conclusion that urinary GDF-15 is not a predictor of stone passage, which is a significant negative finding.
7. Format references uniformly per journal guidelines (e.g., titles in sentence case, consistent DOI presentation).
Comments on the Quality of English LanguageThroughout, there are a few minor grammar mistakes and uncomfortable phrases (such as "Patients who underwent surgical intervention... were excluded"; this could add selection bias).
Consider using "urinary GDF-15" consistently instead of "urine GDF-15".
Author Response
Author Response for Reviewer-1
We sincerely thank Reviewer 1 for the careful evaluation of our manuscript and for the constructive and insightful comments provided. The reviewer’s suggestions greatly helped us improve the clarity, methodological transparency, and overall quality of the manuscript.
In response, we clarified the rationale for the sample size calculation, refined the interpretation of correlation analyses to avoid overstatement, and revised the Discussion section to reduce redundancy and improve structure. The mechanistic description of GDF-15 was condensed and better integrated into the clinical context. We also expanded the Discussion and Conclusion sections to address biomarker specificity, medication-related confounding factors, and to clearly state that urinary GDF-15 is not a predictor of spontaneous stone passage. In addition, the ROC analysis was strengthened by reporting sensitivity and specificity with 95% confidence intervals and by providing a clearer interpretation of the identified cut-off values. Finally, reference formatting, terminology consistency, and minor language issues were corrected throughout the manuscript in accordance with the journal’s guidelines.
We believe that these revisions have substantially improved the manuscript and addressed all concerns raised by the reviewer.
Comment 1: Although a power calculation is referenced (G*Power; d = 0.60; α = 0.05; power = 0.90), the chosen effect size is not justified. Please cite precedent studies or provide a rationale for selecting d = 0.60. Despite the link between serum GDF-15 and CRP being characterized as 'significant' (r = 0.385, p < 0.05), its strength is mild to moderate. This should be reframed to avoid overstatemet. Similarly, there is virtually little connection (r = 0.275) between urine GDF-15 and CRP. The wording should properly acknowledge this.
Response 1: We thank the reviewer for this constructive and insightful comment, which helped us improve both the methodological transparency and the interpretation of our findings. First, the rationale for selecting a moderate effect size (Cohen’s d = 0.60) in the power analysis has been clarified in the Materials and Methods section. This value was chosen based on precedent clinical studies evaluating serum and urinary GDF-15 as renal and inflammatory biomarkers in patient cohorts of comparable size, in which moderate between-group differences were reported. Specifically, studies by Perez-Gomez et al. and Oshita et al. investigated urinary GDF-15 in renal disease populations with sample sizes similar to our cohort and demonstrated effect magnitudes consistent with a moderate effect size. In the absence of ureteral stone–specific GDF-15 data, we therefore selected a conservative moderate effect size to balance statistical rigor and feasibility. Relevant references have been added to support this choice. Second, we fully agree that statistical significance does not necessarily imply a strong biological association. Accordingly, we revised the Results section (Section 3.5) to more accurately reflect the strength of the observed correlations. The association between serum GDF-15 and CRP (r = 0.385) is now described as a statistically significant but mild-to-moderate correlation, while the association between urinary GDF-15 and CRP (r = 0.275) is explicitly characterized as weak, indicating a limited relationship with systemic inflammatory activity. Third, to ensure consistency and avoid overstatement, the corresponding interpretation in the Discussion has been revised. The revised text now emphasizes that, although GDF-15 may partially reflect inflammatory processes associated with ureteral obstruction, the modest correlation coefficients observed—particularly with CRP—indicate that GDF-15 should not be interpreted as a direct surrogate marker of systemic inflammation. These revisions ensure a more balanced, transparent, and statistically appropriate presentation of the data, in line with the reviewer’s recommendations.
Comment 2: The discussion is lengthy and repetitive in places. There are several pages devoted to CT overuse, for example. It should be condensed and rearranged.
Response 2: We sincerely thank the reviewer for this helpful and constructive comment regarding the length and organization of the Discussion section. We appreciate the opportunity to improve the clarity and readability of the manuscript. In response to this comment, we carefully revised the Discussion to reduce redundancy and improve structure. Specifically, the sections addressing computed tomography (CT) use and overuse in the emergency setting were condensed and reorganized into a single, concise paragraph. This revision retains the essential clinical context and supporting literature while avoiding repetitive statements and excessive emphasis on CT-related limitations. We believe that this modification has substantially improved the flow and focus of the Discussion section, making the key messages clearer and more balanced without altering the interpretation of our findings.
Comment 3: Lines 358–367, which describe GDF-15 as a "inflammation-induced central mediator," seem a little pointless. It should be integrated more concisely.
Response 3: We sincerely thank the reviewer for this helpful comment regarding the organization and focus of the Discussion section. We agree that the previous text describing GDF-15 as an “inflammation-induced central mediator of tissue tolerance” was overly detailed and could distract from the main clinical message of the manuscript. In response, we condensed this mechanistic explanation and integrated it into the surrounding discussion rather than presenting it as a standalone paragraph. The revised text now summarizes the concept in a more concise manner and directly links it to our findings by emphasizing that elevated GDF-15 in ureteral stone patients may reflect a stress-responsive, potentially organ-protective response to acute renal stress and inflammation. Importantly, the original supporting references [23–25] were retained while eliminating repetitive or unnecessary mechanistic detail. We believe this revision improves clarity, coherence, and readability of the Discussion without altering the interpretation of our results.
Comment 4: Describe how GDF-15 is raised in a number of systemic illnesses, including cancer, chronic kidney disease (CKD), and cardiovascular disease (CVD). What impact might this have on specificity in actual emergency situations?. Consider how analgesics or MET (alpha-blockers) may affect inflammatory indicators.
Response 4: We sincerely thank the reviewer for this important comment regarding the specificity and clinical interpretation of GDF-15. In response, we have expanded the limitations section of the Discussion to explicitly acknowledge that GDF-15 is elevated in a variety of systemic conditions, including cardiovascular disease, chronic kidney disease, and malignancy, which may limit its specificity in emergency settings. We now emphasize that GDF-15 should be interpreted within the broader clinical context rather than as a standalone diagnostic marker. In addition, we have addressed the potential influence of commonly used treatments in renal colic management, such as analgesics and medical expulsive therapy (alpha-blockers), on inflammatory pathways and systemic stress responses. While these treatments were administered according to standard clinical practice, their possible effects on inflammatory biomarkers are now discussed as a limitation and an area for future investigation.
Comment 5: Concerning the ROC Analysis Presentation, give sensitivity and specificity 95% confidence intervals (CIs) in addition to the area under the curve (AUC). The cut-off point for urinary GDF-15 (438.3 pg/mL) should be discussed.
Response 5: We sincerely thank the reviewer for this valuable comment regarding the ROC analysis. In response, we have expanded the Results section to report sensitivity and specificity values together with their corresponding 95% confidence intervals at the identified cut-off points for both urinary and serum GDF-15. These additions improve the completeness and transparency of the diagnostic performance reporting. In addition, we have revised the Discussion to clarify that the urinary GDF-15 cut-off value (438.3 pg/mL) was data-driven and derived from the current cohort, and should therefore be considered exploratory pending external validation in independent populations.
Comment 6: Declare clearly in the conclusion that urinary GDF-15 is not a predictor of stone passage, which is a significant negative finding.
Response 6: We sincerely thank the reviewer for highlighting this important point. In response, we have revised the Conclusion section to explicitly state that urinary GDF-15 was not a predictor of spontaneous stone passage in our study. We now emphasize this finding as a meaningful negative result, reflecting the multifactorial nature of stone passage and clarifying that the clinical utility of urinary GDF-15 in our cohort is primarily diagnostic rather than prognostic.
Comment 7: Format references uniformly per journal guidelines (e.g., titles in sentence case, consistent DOI presentation).
Response 7: We thank the reviewer for this helpful comment. In response, all references have been carefully reviewed and reformatted to ensure full compliance with the journal’s guidelines.
Comment 8: Throughout, there are a few minor grammar mistakes and uncomfortable phrases (such as "Patients who underwent surgical intervention... were excluded"; this could add selection bias). Consider using "urinary GDF-15" consistently instead of "urine GDF-15".
Response 8: We thank the reviewer for this helpful comment regarding language clarity and consistency. In response, the manuscript was carefully revised to correct minor grammatical issues and improve phrasing throughout the text. Specifically, sentences that could imply unintended selection bias—particularly those describing patient exclusion during follow-up—were reworded to provide clearer methodological justification. Redundant or repetitive statements were removed to enhance readability. In addition, terminology was standardized, and the expression “urinary GDF-15” is now used consistently throughout the manuscript.
Reviewer 2 Report
Comments and Suggestions for Authors
This study is devoted to investigating the role of GDF-15—a cytokine from the transforming growth factor-beta family—in the progression of obstructive nephropathy caused by ureteral stones. The authors propose using serum and urinary GDF-15 as a biomarker for this pathology, alongside traditional diagnostic methods for urolithiasis accompanied by renal colic. The study is well-designed, the sample size is justified, the accompanying clinical diagnostics of patients are described in detail, and the statistical analysis of the obtained data is conducted correctly. There are minor comments for the authors:
-
Are there any data on the stability of GDF-15 in biological fluids? Could it be subject to proteolytic degradation upon freezing, which might have affected the relatively weak correlations between GDF-15 levels and inflammatory markers in the correlation matrix shown in Fig. 4?
-
It might be useful to add information on what is known regarding regarding the dynamics of GDF-15 reduction in the blood as the inflammatory process gradually subsides during recovery.
-
Is it possible to assess whether GDF-15 levels differ between patients with ureteral obstruction due to stones and patients whose stones have already passed through the ureter but who still have a residual inflammatory process?
Author Response
Authors Response for Reviewer-2
We sincerely thank Reviewer 2 for the careful evaluation of our manuscript and for the constructive and helpful comments. In response, we revised the Discussion and Limitations sections to clarify issues related to the biological interpretation of GDF-15, including its behavior during recovery from acute inflammatory conditions, pre-analytical considerations, and the distinction between active ureteral obstruction and post-obstructive residual inflammation. These revisions were made to improve clarity and better contextualize our findings within existing literature.
Reviewer-2
This study is devoted to investigating the role of GDF-15—a cytokine from the transforming growth factor-beta family—in the progression of obstructive nephropathy caused by ureteral stones. The authors propose using serum and urinary GDF-15 as a biomarker for this pathology, alongside traditional diagnostic methods for urolithiasis accompanied by renal colic. The study is well-designed, the sample size is justified, the accompanying clinical diagnostics of patients are described in detail, and the statistical analysis of the obtained data is conducted correctly. There are minor comments for the authors:
Comment 1: Are there any data on the stability of GDF-15 in biological fluids? Could it be subject to proteolytic degradation upon freezing, which might have affected the relatively weak correlations between GDF-15 levels and inflammatory markers in the correlation matrix shown in Fig. 4?
Response 1: We thank the reviewer for this important methodological comment. In response, we have added a brief statement to the Limitations section noting that, although standardized sample handling and storage procedures were applied to minimize pre-analytical variability, freshly processed samples may provide more accurate GDF-15 measurements. We also clarify that frozen sample analysis represents a practical and widely accepted approach in biomarker studies, as GDF-15 is not routinely measured in acute clinical settings.
Comment 2: It might be useful to add information on what is known regarding regarding the dynamics of GDF-15 reduction in the blood as the inflammatory process gradually subsides during recovery.
Response 2: We thank the reviewer for this valuable comment. In response, we have expanded the Discussion to briefly address the temporal dynamics of GDF-15 during recovery from acute inflammatory conditions. We now reference our previous prospective study in urosepsis patients, which demonstrated a significant decline in serum GDF-15 levels during clinical recovery, supporting the concept that GDF-15 reflects dynamic changes related to resolution of systemic inflammatory stress. We also clarify that longitudinal changes could not be assessed in the present cross-sectional study.
Comment 3: Is it possible to assess whether GDF-15 levels differ between patients with ureteral obstruction due to stones and patients whose stones have already passed through the ureter but who still have a residual inflammatory process?
Response 3: We thank the reviewer for this important and clinically relevant question. In the present study, GDF-15 levels were measured only at the time of initial presentation in patients with radiologically confirmed ureteral obstruction. Patients who had already passed their stones but had residual inflammatory findings were not included as a separate group, and post-passage GDF-15 measurements were not performed. Therefore, differences between active obstruction and post-obstructive residual inflammation could not be assessed. This limitation has now been clarified in the Discussion and should be addressed in future longitudinal studies with serial biomarker measurements.